# A Non-asymptotic Analysis of Non-parametric Temporal-Difference Learning

**Eloïse Berthier**
Inria & Ecole Normale Supérieure
PSL Research University, Paris, France
`eloise.berthier@inria.fr`

**Ziad Kobeissi**
Institut Louis Bachelier
Inria Paris
`ziad.kobeissi@inria.fr`

**Francis Bach**
Inria & Ecole Normale Supérieure
PSL Research University, Paris, France
`francis.bach@inria.fr`

## Abstract

Temporal-difference learning is a popular algorithm for policy evaluation. In this paper, we study the convergence of the regularized non-parametric TD(0) algorithm, in both the independent and Markovian observation settings. In particular, when TD is performed in a universal reproducing kernel Hilbert space (RKHS), we prove convergence of the averaged iterates to the optimal value function, even when it does not belong to the RKHS. We provide explicit convergence rates that depend on a source condition relating the regularity of the optimal value function to the RKHS. We illustrate this convergence numerically on a simple continuous-state Markov reward process.

## 1   Introduction

One of the main ingredients of reinforcement learning (RL) is the ability to estimate the long-term effect on future rewards of employing a given policy. This building block, known as policy evaluation, already contains crucial features of more complex RL algorithms, such as SARSA or Q-learning [64]. Temporal-difference learning (TD), proposed by [62], is among the simplest algorithms for policy evaluation. The estimation of the performance of the policy is made through a value function. It is updated *online*, after each new observation of a couple composed of a state transition and a reward.

Although the formulation of TD is quite natural, its theoretical analysis has proved more challenging, as it combines two difficulties. The first one is that TD *bootstraps*, in the sense that it uses its previous – possibly inaccurate – predictions to correct its next predictions, because it does not have access to a fixed ground truth. The second difficulty is that the observations are produced along a trajectory following a fixed policy (*on-policy*), hence they are correlated, which calls for more involved stochastic approximation tools compared to independent identically distributed (*i.i.d.*) samples. Moreover, using *off-policy* samples, produced by a different policy than the one being evaluated, can make the algorithm diverge [16]. Off-policy sampling is out of our scope in this paper.

A third element which is not inherent to TD further complicates the plot: function approximation. While TD was originally proposed in a tabular setting, its large-scale applicability has been greatly extended by its combination with parametric function approximation [17]. This enables the use of any linear or non-linear function approximation method to model the value function, including neural networks. However, one can exhibit unstable diverging behaviors of TD even with simple non-linear approximation schemes [66]. This combination of difficulties (even with linear function approxima-

36th Conference on Neural Information Processing Systems (NeurIPS 2022).

tion) has been coined the "deadly triad" by [63]. We argue that convergence can be obtained even with non-linear function approximation, by making use of the non-parametric formalism of reproducing kernel Hilbert spaces (RKHS), involving linear approximation in infinite-dimension. Studying this case could bring us closer to understanding what happens with other universal approximators used in practice, like neural networks.

## 1.1 Contributions

We study the policy evaluation algorithm TD(0) in the non-parametric case, first when the observations are sampled *i.i.d.* from the invariant distribution of the Markov chain resulting from the evaluated policy, and then when they are collected from a trajectory of the Markov chain with geometric mixing. In that sense we follow a similar outline as the analysis of [10] which is dedicated to the linear case.

The non-parametric formulation of TD closes the gap between the original tabular formulation and the parametric formulation which involves semi-gradients. It allows the use of classical tools and theory from kernel methods [22]. In particular, we highlight the central role of infinite-dimensional covariance operators [5, 2] which already appear in the analysis of other non-parametric algorithms, like least-squares regression. We study a regularized variant of TD, a widely used way of dealing with misspecification in regression. Importantly, when the regularized TD approximation is run on an infinite-dimensional RKHS which is dense in the space of square-integrable functions, then there is no approximation error and the algorithm converges to the true value function. More precisely, we provide a proof of convergence in expectation of TD without approximation error, even when the true value function does not belong to the RKHS, under a weaker source condition. Furthermore, we give non-asymptotic convergence rates related to this source condition, which measures the regularity of the true value function relative to the RKHS, *e.g.*, its smoothness if the RKHS is a Sobolev space [51].

Note that using a universal kernel [48] to obtain convergence of TD to the true value function is also interesting from a theoretical point of view. Indeed it exempts us from a possibly tedious study of the approximation (or projection) error on a given basis, and simply removes an error term which in general scales linearly with the horizon of the Markov reward process [49, 70].

In the rest of this section, we review the related literature. In Sec. 2, we present the algorithm, along with generic results and notations. In Sec. 3, we analyze a simplified version of the algorithm, namely population TD in continuous time. This allows to catch the main features of the analysis, while postponing the technicalities related to stochastic approximation. Sec. 4 is dedicated to the analysis of non-parametric TD with *i.i.d.* observations, while Sec. 5 consists in a similar analysis for correlated observations sampled from a geometrically mixing Markov chain. Finally, in Sec. 6, we present simple numerical simulations illustrating the convergence results and the role the main parameters.

## 1.2 Related literature

**Temporal-difference learning.**    The TD algorithm was introduced in its tabular version by [62], with a first convergence result for linearly independent features, later extended to dependent features by [27]. Further stochastic approximation results were proposed by [41] for the tabular case, and by [58] for the linear approximation case. An exact analysis of the behavior of tabular TD was recently carried out by [40], using the framework of Markov jump linear systems. [66] provided a thorough asymptotic analysis of TD with linear function approximation, while failure cases were already known [4]. A non-asymptotic analysis was later proposed by [45] in the *i.i.d.* sampling case with constant step size, concurrently to another approach extending to Markov sampling by [10]. Other problem-dependent bounds for linear TD were derived around the same period [26, 60], along with an analysis of variance-reduced TD [44, 69]. All of the analyses mentioned above focus either on the tabular or on the linear *parametric* TD algorithm. A recent work by [33] deals with the batch counterpart of non-parametric TD, namely the least-squares TD algorithm (LSTD), but they rather focus on the analysis of the statistical estimation error. Importantly, LSTD only requires offline computations and is not related to stochastic approximation. Most closely related to our work is the non-parametric regularized TD setting studied by [43]. However, their analysis is limited to the case where the optimal value function belongs to the RKHS. This is not sufficient to get rid of the approximation error term. Also, we will show later that regularization is not necessary in this case. Furthermore, their analysis is restricted to the *i.i.d.* setting, for which we will require fewer regularity assumptions. Finally, let us mention the recent work by [18] concerning TD with function

approximation using a one-hidden layer neural network with finite-width, called "neural TD". Since finite-width neural networks are not universal approximators, there is an approximation error, which vanishes in the infinite-width limit if the value function belongs to a particular function space.

**Kernel methods in RL.** To tackle large-dimensional problems, kernel methods have been combined with various RL algorithms, including approximate dynamic programming [53, 11, 6, 38], policy evaluation [25], policy iteration [36], LSTD [33], the linear programming formulation of RL [29], upper confidence bound [32], or fitted Q-iteration [47]. Such kernel methods often come along with practical ways to reduce the computational complexity that grows with the number of observed transitions and rewards [7, 43].

**Stochastic approximation.** The analysis of TD requires tools from stochastic approximation [8], among which the ODE method [13]. Such tools are primarily designed for finite-dimensional problems. Stochastic gradient descent (SGD) [15] is a specific instance of stochastic approximation that has received extensive attention for supervised learning. In particular, the role of regularization of SGD for least-squares regression has been studied [19, 24], as well as the effect of of sampling data from a Markov chain [50]. Finally, we use a formalism which is close to the analyses [31, 54, 9] of non-parametric SGD for least squares regression.

## 2 Problem formulation and generic results

### 2.1 The non-parametric TD(0) algorithm

We consider a Markov reward process (MRP), *i.e.*, a Markov chain with a reward associated to each state. This is what results from keeping the policy fixed in a Markov decision process (MDP) for policy evaluation. We consider MRPs in discrete-time, not necessarily with a countable state space $\mathcal{X}$. Specifically, we use the formalism of Markov chains on a measurable state space, also called Harris chains, which unifies discrete- and continuous-state Markov chains. Formally, let $\mathcal{X} \subset \mathbb{R}^d$ a measurable set associated with the $\sigma$-algebra $\mathcal{A}$ of Lebesgue measurable sets. Let $(x_n)_{n \geq 1}$ a time-homogeneous Markov chain with Markov kernel $\kappa$. A Markov kernel [56, 42] is a mapping $\kappa : \mathcal{X} \times \mathcal{A} \to [0, 1]$ that has the following two properties:

1. for every $x \in \mathcal{X}$, $\kappa(x, \cdot)$ is a probability measure on $\mathcal{A}$, and
2. for every $A \in \mathcal{A}$, $\kappa(\cdot, A)$ is $\mathcal{A}$-measurable.

If $\mathcal{X}$ is a countable set, $\kappa$ is represented by a transition matrix $Q$ such that $Q_{i,j} := \mathbb{P}(j|i) = \kappa(i, \{j\})$, for any $i, j \in \mathcal{X}$.

We define a reward function $r : \mathcal{X} \to \mathbb{R}$ uniformly bounded by $R < \infty$, and a discount factor $\gamma \in [0, 1)$. The aim of policy evaluation is to compute the value function of the MRP:

$$\forall x \in \mathcal{X}, \quad V^*(x) = \mathbb{E}\Big[ \sum_{n=0}^{+\infty} \gamma^n r(x_n) \,\Big|\, x_0 = x \Big], \tag{1}$$

where the $(x_n)_{n \geq 1}$ are drawn from the Markov chain. A probability distribution $p : \mathcal{A} \to \mathbb{R}$ is a stationary distribution for $\kappa$ if for all $A \in \mathcal{A}$, $p(A) = \int_{\mathcal{X}} \kappa(x, A)p(dx)$. The existence and uniqueness of a stationary distribution $p$, along with the convergence of the Markov chain to $p$ in total variation, is ensured by ergodicity conditions. A sufficient condition is that the Markov chain is Harris ergodic, *i.e.*, it has a regeneration set, and is aperiodic and positively recurrent (see [1] and [35] for a complete exposition of Harris chains). For discrete-state Markov chains, ergodicity conditions can be expressed somewhat more simply, and any aperiodic and positive recurrent Markov chain has a unique invariant distribution. Throughout this paper, we assume that $p$ is the unique invariant distribution of the Markov chain, and that it has full support on $\mathcal{X}$. Only in Sec. 5, we will in addition assume that the Markov chain is geometrically mixing.

We define $L^2(p)$, the set of squared integrable functions $f : \mathcal{X} \to \mathbb{R}$ with respect to $p$, with the norm $\|f\|_{L^2(p)}^2 = \int_{\mathcal{X}} f(x)^2 p(dx) < +\infty$. We also consider a reproducing kernel Hilbert space $\mathcal{H}$ of $\mathcal{A}$-measurable functions, associated to a positive-definite kernel $K : \mathcal{X} \times \mathcal{X} \to \mathbb{R}$. For all $x \in \mathcal{X}$, we use the notation $\Phi(x) := K(x, \cdot)$ for the mapping of $x$ in $\mathcal{H}$, and $\langle \cdot, \cdot \rangle_{\mathcal{H}}$ for the inner product in $\mathcal{H}$

(we sometimes drop the index). We assume that $M_{\mathcal{H}} := \sup_{x \in \mathcal{X}} K(x,x)$ is finite, which implies that $\mathcal{H} \subset L^2(p)$. More precisely, the $\mathcal{H}$-norm controls the $L^2(p)$-norm: any sequence converging in $\mathcal{H}$ thus converges in $L^2(p)$. Indeed, if $f \in \mathcal{H}$:

$$\|f\|_{L^2(p)}^2 = \int f(x)^2 dp(x) = \int \langle f, \Phi(x)\rangle_{\mathcal{H}}^2 dp(x) \le \|f\|_{\mathcal{H}}^2 \int \|\Phi(x)\|_{\mathcal{H}}^2 dp(x) \le M_{\mathcal{H}} \|f\|_{\mathcal{H}}^2. \quad (2)$$

We also assume that $r \in L^2(p)$. The non-parametric TD(0) algorithm in the RKHS $\mathcal{H}$ is defined as follows [53, 43]. Draw a sequence $(x_n)_{n \ge 0}$ according to the Markov chain with initial distribution $p$, and collect the corresponding rewards $(r(x_n))_{n \ge 0}$. Define a sequence of non-negative step sizes $(\rho_n)_{n \ge 1}$. We build recursively a sequence of approximate value functions $(V_n)_{n \ge 0}$ in $L^2(p)$. Throughout the paper, we take $V_0 = 0$ for simplicity, but note that all the results can be adapted to the case $V_0 \in \mathcal{H}$. For $n \ge 1$:

$$\forall y \in \mathcal{X}, \quad V_n(y) = V_{n-1}(y) + \rho_n \Big[ r(x_n) + \gamma V_{n-1}(x_n') - V_{n-1}(x_n) \Big] K(x_n, y), \quad (3)$$

where $x_n' := x_{n+1}$. The term in brackets is called a temporal-difference. Equivalently, in the RKHS:

$$V_n = V_{n-1} + \rho_n \Big[ r(x_n) + \gamma V_{n-1}(x_n') - V_{n-1}(x_n) \Big] \Phi(x_n). \quad (4)$$

This update has a running time complexity of $O(n^2)$, which can be improved to $O(n)$, *e.g.* using Nyström approximation or random features [39]. More details on the implementation are given in App. B.2. This non-parametric formulation is a natural extension of the tabular TD algorithm. Indeed, if $\mathcal{X}$ is a countable set and $K(x,y) = \mathbf{1}_{x=y}$ is a Dirac kernel – a valid positive-definite kernel – then we exactly recover tabular TD: the update rule (3) becomes, after observing a transition $(i, i', r_i) := (x_n, x_n', r(x_n))$:

$$V_n(i) = V_{n-1}(i) + \rho_n \Big[ r_i + \gamma V_{n-1}(i') - V_{n-1}(i) \Big], \quad \text{and} \quad \forall j \ne i, \ V_n(j) = V_{n-1}(j). \quad (5)$$

This also covers the *semi-gradient* formulation of TD for linear function approximation [64]. Suppose $\mathcal{H}$ has finite dimension $d$, then $V_n$ can be identified to $\xi_n \in \mathbb{R}^d$, and we are searching for an approximation of the form $V_n(x) = \xi_n^\top \Phi(x)$. Then (4) becomes:

$$\xi_n = \xi_{n-1} + \rho_n \Big[ r(x_n) + \gamma V_{n-1}(x_n') - V_{n-1}(x_n) \Big] \nabla_\xi V_n(x_n). \quad (6)$$

Since $V_0 \in \mathcal{H}$, all the iterates $V_n$ are in the RKHS, in particular $V_n \in \mathrm{span}\{\Phi(x_k)\}_{1 \le k \le n}$. Consequently, if the sequence $(V_n)$ converges in the topology induced by the $L^2(p)$-norm, it converges in $\overline{\mathcal{H}}$, the closure of $\mathcal{H}$ with respect to the $L^2(p)$-norm. In particular, for a dense RKHS and because $p$ has full support on $\mathcal{X}$, $\overline{\mathcal{H}} = L^2(p)$, but in general it only holds that $\overline{\mathcal{H}} \subset L^2(p)$.

To understand the behavior of the algorithm, we will first consider the *population* version (also called *mean-path* in [10]) of the algorithm: set $V_0 = 0$ and for $n \ge 1$:

$$V_n = V_{n-1} + \rho_n \mathbb{E}_{(x,x') \sim q} \left[ \left( r(x) + \gamma V_{n-1}(x') - V_{n-1}(x) \right) \Phi(x) \right], \quad (7)$$

where the expectation is taken with respect to $q(dx, dx') := p(dx)\kappa(x, dx')$. Since $V_{n-1} \in \mathcal{H}$, the reproducing property holds: $V_{n-1}(x) = \langle V_{n-1}, \Phi(x)\rangle_{\mathcal{H}}$. Hence the update is affine and reads: $V_n = V_{n-1} + \rho_n(AV_{n-1} + b)$, with $A := \mathbb{E}_q [\gamma \Phi(x) \otimes \Phi(x') - \Phi(x) \otimes \Phi(x)]$ and $b := \mathbb{E}_p [r(x)\Phi(x)]$, where $\otimes$ denotes the outer product in $\mathcal{H}$ defined by $g \otimes h : f \mapsto \langle f, h\rangle_{\mathcal{H}} g$.

## 2.2 Covariance operators

Assume that the expectations $\Sigma := \mathbb{E}_p[\Phi(x) \otimes \Phi(x)]$ and $\Sigma_1 := \mathbb{E}_q[\Phi(x) \otimes \Phi(x')]$ are well-defined. $\Sigma$ and $\Sigma_1$ are the uncentered auto-covariance operators of order 0 and 1 of the Markov process $(x_n)_{n \ge 1}$, under the invariant distribution $p$. They are operators from $\mathcal{H}$ to $\mathcal{H}$, such that, for all $f, g \in \mathcal{H}$, using the reproducing property:

$$\begin{aligned}
\mathbb{E}_p[f(x)g(x)] &= \mathbb{E}_p[\langle f, \Phi(x)\rangle_{\mathcal{H}} \langle g, \Phi(x)\rangle_{\mathcal{H}}] = \langle f, \mathbb{E}_p[\langle g, \Phi(x)\rangle_{\mathcal{H}} \Phi(x)]\rangle_{\mathcal{H}} = \langle f, \Sigma g\rangle_{\mathcal{H}} \\
\mathbb{E}_q[f(x)g(x')] &= \mathbb{E}_q[\langle f, \Phi(x)\rangle_{\mathcal{H}} \langle g, \Phi(x')\rangle_{\mathcal{H}}] = \langle f, \mathbb{E}_p[\langle g, \Phi(x')\rangle_{\mathcal{H}} \Phi(x)]\rangle_{\mathcal{H}} = \langle f, \Sigma_1 g\rangle_{\mathcal{H}}.
\end{aligned} \quad (8)$$

In particular, for all $y \in \mathcal{X}$ and $f \in \mathcal{H}$, $(\Sigma f)(y) = \langle \Phi(y), \Sigma f \rangle_{\mathcal{H}} = \mathbb{E}_p[f(x)K(x,y)]$ and similarly, $(\Sigma_1 f)(y) = \mathbb{E}_q[f(x')K(x,y)]$. Following [31], $\Sigma$ and $\Sigma_1$ can therefore be extended to operators $\Sigma^e$ and $\Sigma_1^e$ from $L^2(p)$ to $L^2(p)$ defined by:

$$\Sigma^e : f \mapsto \int_{\mathcal{X}} f(x)\Phi(x)p(dx), \text{ such that } \forall y \in \mathcal{X}, \ (\Sigma^e f)(y) = \mathbb{E}_p[f(x)K(x,y)]$$
$$\Sigma_1^e : f \mapsto \iint_{\mathcal{X}^2} f(x')\Phi(x)q(dx, dx'), \text{ such that } \forall y \in \mathcal{X}, \ (\Sigma_1^e f)(y) = \mathbb{E}_q[f(x')K(x,y)]. \tag{9}$$

These two operators are the building blocks of the TD iteration (7). In particular, $A = \gamma \Sigma_1 - \Sigma$ and $b = \Sigma^e r$, the latter being valid for $r \in L^2(p)$. With a slight abuse of notation, we denote simply as $\Sigma$, $\Sigma_1$ the extended operators. Furthermore [31], $\text{Im}(\Sigma) \subset \mathcal{H}$ and $\Sigma^{1/2}$ is an isometry from $L^2(p)$ to $\mathcal{H}$:

$$\forall f \in \overline{\mathcal{H}}, \quad \|f\|_{L^2(p)} = \|\Sigma^{1/2} f\|_{\mathcal{H}}. \tag{10}$$

The fact that $p$ is a stationary distribution for $\kappa$ implies a particular constraint linking $\Sigma$ and $\Sigma_1$:

**Lemma 1.** *There exists a unique bounded linear operator $\tilde{\Sigma}_1 : \mathcal{H} \to \mathcal{H}$ such that $\Sigma_1 = \Sigma^{1/2} \tilde{\Sigma}_1 \Sigma^{1/2}$ on $\overline{\mathcal{H}}$, and $\|\tilde{\Sigma}_1\|_{\mathrm{op}} \le 1$ ($\|\cdot\|_{\mathrm{op}}$ is the $\mathcal{H}$-operator norm).*

This results from an application of [5, Thm. 1], valid on $\mathcal{H}$ and extended by continuity to $\overline{\mathcal{H}}$. See also [37] for an exposition of cross-covariance operators specifically in an RKHS. In finite dimension, this is retrieved with generic results on positive semi-definite (PSD) matrices. Specifically, if $\mathcal{H} \subset \mathbb{R}^m$, the uncentered covariance matrix of the random variable $(\Phi(x), \Phi(x'))$, when $(x, x') \sim q$ is:

$$\begin{pmatrix} \Sigma & \Sigma_1 \\ \Sigma_1^\top & \Sigma \end{pmatrix} \succeq 0.$$

Using a classical condition on block matrices [12, Prop. 1.3.2], this matrix is PSD if and only if there exists a matrix $\tilde{\Sigma}_1$ such that $\|\tilde{\Sigma}_1\|_{\mathrm{op}} \le 1$ and $\Sigma_1 = \Sigma^{1/2}\tilde{\Sigma}_1 \Sigma^{1/2}$ ($\|\cdot\|_{\mathrm{op}}$ is also the spectral norm in this case). This corresponds to the fact that the Schur complement of a PSD block matrix is also PSD.

**Assumptions on $\Sigma$ and $V^*$.** We assume that $x \mapsto K(x, x)$ is uniformly bounded by $M_{\mathcal{H}}$. Therefore, the eigenvalues of $\Sigma$ are upper-bounded. However, unlike [66] and [10], we do not assume them to be lower-bounded, *i.e.*, $\Sigma \succeq 0$ is not invertible in general. We will formulate our convergence results for two sets of assumptions. The first one recovers known results from [10] for linear function approximation. The second one assumes that $V^*$ verifies a *source condition* [30, Chap. 1]:

(A1) $V^* \in \mathcal{H}$, $\mathcal{H}$ is finite-dimensional and $\Sigma$ has full-rank;
(A2) $V^* \in \Sigma^{\theta/2}(\mathcal{H})$ for some $\theta \in (-1, 1]$ (and consequently, $\|\Sigma^{-\theta/2} V^*\|_{\mathcal{H}} < +\infty$), and $\overline{\mathcal{H}} = L^2(p)$ (*i.e.*, $K$ is a universal kernel).

In (A1), $\mathcal{H}$ is finite-dimensional because $\Sigma$ cannot be simultaneously compact ($x \mapsto K(x, x)$ being uniformly bounded) and invertible in infinite-dimension [21]. Recalling the isometry property (10), the case $\theta = -1$ always holds in (A2) because $V^* \in L^2(p)$ (which we prove in the next subsection). The case $\theta = 0$ is equivalent to $V^* \in \mathcal{H}$. For $\theta > 0$, it must be interpreted as: $\|\Sigma^{-\theta/2} V^*\|_{\mathcal{H}}^2 := \inf\{\|V\|_{\mathcal{H}}^2 \mid V \text{ s.t. } V^* = \Sigma^{\theta/2} V\}$, with $\|\Sigma^{-\theta/2} V^*\|_{\mathcal{H}} = +\infty$ if $V^* \notin \Sigma^{\theta/2}(\mathcal{H})$. Using a universal approximation removes the need for a projection operator on $\overline{\mathcal{H}}$, as typically used for finite-dimensional function approximation, and hence there will be no projection error [66].

### 2.3 Non-expansiveness of the Bellman operator

It is known that the value function $V^*$ of the MRP is a fixed point of the Bellman operator $T$. We define two operators $P$ and $T : L^2(p) \to L^2(p)$ by, for $V \in L^2(p)$, $PV(x) = \mathbb{E}_{x' \sim \kappa(x, \cdot)} V(x')$ and $TV(x) = r(x) + \gamma PV(x)$. Both operators can be expressed in terms of $\Sigma$ and $\Sigma_1$. For $V \in L^2(p)$:

$$\begin{cases} \Sigma PV = \mathbb{E}_p[\Phi(x)(PV)(x)] = \mathbb{E}_q[\Phi(x)V(x')] = \Sigma_1 V \\ \Sigma TV = \Sigma r + \gamma \Sigma_1 V. \end{cases} \tag{11}$$

**Lemma 2.** *For any $V \in L^2(p)$: $\|PV\|_{L^2(p)} \le \|V\|_{L^2(p)}$.*

This is a direct reformulation of [66, Lemma 1], the proof of which is given in App. A.1. As stressed by [66], this strongly relies on the fact that $p$ is a stationary distribution of the Markov chain. It implies that $T$ is a $\gamma$-contraction mapping on $L^2(p)$ and has as unique fixed point $V^*$. One can check that if $\Sigma$ is non-singular, Lemma 2 is exactly equivalent to $\|\Sigma^{-1/2}\Sigma_1\Sigma^{-1/2}\|_{\mathrm{op}} \leq 1$, that is, Lemma 1. Moreover, using Lemma 2, we obtain $\|V^*\|_{L^2(p)} \leq \|r\|_{L^2(p)}/(1-\gamma)$ and $V^* \in L^2(p)$.

# 3 Analysis of a continuous-time version of the population TD algorithm

Before considering regularized TD with stochastic samples, we look at simplified versions of the algorithm that momentarily remove the difficulties related to stochastic approximation. Specifically, we consider the population version of TD to capture a "mean" behavior, and a continuous-time algorithm to avoid choosing step sizes. Instead, we focus on the role of the regularization parameter.

## 3.1 Existence of a fixed-point for regularized TD

For $\lambda \geq 0$, let us consider the regularized population recursion:

$$V_n = V_{n-1} + \rho_n(\Sigma r + (\gamma\Sigma_1 - \Sigma - \lambda I)V_{n-1}). \tag{12}$$

If the TD iterations converge, their limit will be a solution of the *regularized* fixed-point equation:

$$\Sigma r + (\gamma\Sigma_1 - \Sigma - \lambda I)V = 0. \tag{13}$$

**Proposition 1.** *If $\lambda > 0$, $\gamma\Sigma_1 - \Sigma - \lambda I$ is non-singular on $\mathcal{H}$ and equation (13) admits a unique solution $V_\lambda^*$ in $L^2(p)$, defined by $V_\lambda^* = (\gamma\Sigma_1 - \Sigma - \lambda I)^{-1}\Sigma r$. Furthermore, $V_\lambda^* \in \mathcal{H}$ and:*

$$\|V_\lambda^*\|_{\mathcal{H}} \leq \frac{\|\Sigma r\|_{\mathcal{H}}}{\lambda} \leq \frac{\sqrt{M_{\mathcal{H}}}\|r\|_{L^2(p)}}{\lambda}. \tag{14}$$

The proof is in App. A.2. Hence, for $\lambda > 0$, the $\mathcal{H}$-norm of $V_\lambda^*$ is always bounded, unlike $\|V^*\|_{\mathcal{H}}$.

## 3.2 Convergence of the regularized fixed point to the optimal value function

Recalling that $V^* \in L^2(p)$, it satisfies the relation $TV^* = V^*$, implying that $\Sigma TV^* = \Sigma V^*$, *i.e.*, $\Sigma r + (\gamma\Sigma_1 - \Sigma)V^* = 0$. This *unregularized* fixed point equation possibly has other solutions, but if $K$ is a universal kernel, as assumed by (A2), then $\Sigma$ is injective [61] and $V^*$ is the unique solution. Let us recall that (A2) does not imply that $V^*$ has a bounded $\mathcal{H}$-norm. However, we can control the $L^2(p)$-norm of $V_\lambda^* - V^*$ when $\lambda$ is small using the *source condition* (A2).

**Proposition 2.** *Assume that $\lambda > 0$ and assumption (A2). Then:*

$$\|V_\lambda^* - V^*\|_{L^2(p)}^2 \leq \frac{\lambda^{\theta+1}}{(1-\gamma)^2}\|\Sigma^{-\theta/2}V^*\|_{\mathcal{H}}^2. \tag{15}$$

The proof in App. A.2 is inspired by similar results [19, 24] in the context of ridge regression (recovered for $\gamma = 0$). Note that only $\|V_\lambda^* - V^*\|_{L^2(p)}$ is controlled, not $\|V_\lambda^* - V^*\|_{\mathcal{H}}$. Consequently, we obtain the convergence of $V_\lambda^*$ to $V^*$ in $L^2(p)$-norm when $\lambda \to 0$: the higher $\theta$ is, the faster the rate of convergence. For universal Mercer kernels [23], if we drop the source condition (A2), using only the fact that $V^* \in L^2(p)$ – corresponding to $\theta = -1$ in (A2) – we can still prove that $V_\lambda^*$ converges to $V^*$ in $L^2(p)$-norm when $\lambda \to 0$, but without an explicit rate (see App. A.2, Cor. 1).

## 3.3 Convergence of continuous-time population TD

Following the ordinary differential equation (ODE) method [13], we study the continuous-time counterpart of the population iteration (12). At least formally, this consists in defining $\widetilde{V}_t = V_{n(t)}$ for $t$ and $n(t)$ satisfying $t = \sum_{i=1}^{n(t)}\rho_i$, and letting $\rho_i$ tend to 0 for any $i \geq 1$, where $V_{n(t)}$ is defined by recursion using (12). With a slight abuse of notation, we use the notation $V_t$ instead of $\widetilde{V}_t$. We then obtain the following ODE in $\mathcal{H}$: $V_0 = 0$ and for $t \geq 0$:

$$\frac{dV_t}{dt} = (A - \lambda I)V_t + b. \tag{16}$$

We can exhibit a Lyapunov function for this dynamical system, see [59]. This implies that $V_t$ converges to $V_\lambda^*$ when $t$ tends to infinity, where $V_\lambda^*$ is defined in Prop. 1. More precisely, for $\beta \in \{-1, 0\}$, we define $W^\beta$, the Lyapunov function, by $W^\beta(t) := \|\Sigma^{-\beta/2}(V_t - V_\lambda^*)\|_{\mathcal{H}}^2$ (please note that $\beta$'s role in $W^\beta$ is an index, not a power). $W^0(t)$ strictly decreases with $t$ as follows:

**Lemma 3** (Descent Lemma). *For $\lambda > 0$, for all $t \geq 0$, the following holds:*

$$\frac{dW^0(t)}{dt} \leq -2(1-\gamma)W^{-1}(t) - 2\lambda W^0(t), \tag{17}$$

The proof (see App. A.2) mainly relies on the contraction property of the Bellman operator as expressed in Lemma 2. We can then deduce the convergence of the ODE (16) to $V_\lambda^*$.

**Proposition 3.** *Under assumption (A1), the solution $V_t$ of the ODE (16) with $\lambda = 0$ is such that:*

$$\text{For } T > 0, \quad \|\overline{V}_T - V^*\|_{L^2(p)}^2 \leq \frac{1}{2(1-\gamma)}\frac{\|V^*\|_{\mathcal{H}}^2}{T}, \tag{18}$$

*where $\overline{V}_T$ is the Polyak-Ruppert average [55] of $V_t$, defined by $\overline{V}_T := \frac{1}{T}\int_0^T V_t dt$.*

*Under assumption (A2), the solution $V_t$ of the ODE (16) with $\lambda > 0$ is such that:*

$$\text{For } T \geq 0, \quad \|V_T - V_\lambda^*\|_{\mathcal{H}}^2 \leq \|V_\lambda^*\|_{\mathcal{H}}^2 e^{-2\lambda T}. \tag{19}$$

Under (A1), we recover the same $O(1/T)$ convergence rate as [10]. We focus on (A2), where we get a fast convergence to $V_\lambda^*$ in $\mathcal{H}$-norm (stronger than $L^2(p)$). However, we are rather interested in convergence to $V^*$. Prop. 2 quantifies how far $V_\lambda^*$ is from $V^*$. Indeed, the error decomposes as:

$$\|V_T - V^*\|_{L^2(p)}^2 \leq 2M_{\mathcal{H}}\|V_T - V_\lambda^*\|_{\mathcal{H}}^2 + 2\|V_\lambda^* - V^*\|_{L^2(p)}^2. \tag{20}$$

Combining Propositions 1, 2, 3 shows a trade-off on $\lambda$: $\|V_T - V^*\|_{L^2(p)}^2 = O\left(e^{-2\lambda T}/\lambda^2 + \lambda^{\theta+1}\right)$. Taking $\lambda = (3+\theta)\log T/(2T)$ balances the terms up to logarithmic factors: $\|V_T - V^*\|_{L^2(p)}^2 = \tilde{O}\left(T^{-1-\theta}\right)$ (where $\tilde{O}(g(n)) := O(g(n)\log(n)^\ell)$, for some $\ell \in \mathbb{R}$). In particular, for $\theta = 0$, *i.e.*, $V^* \in \mathcal{H}$, we recover a convergence rate $\tilde{O}(1/T)$: up to logarithmic factors, it is the same as the unregularized case with averaging, assuming (A1). In this case, regularization brings no benefits.

## 4 Stochastic TD with *i.i.d.* sampling

We now consider stochastic TD iterations (4), where the couples $(x_n, x_n')_{n\geq 1}$ are sampled *i.i.d.* from the distribution $q(dx, dx') = p(dx)\kappa(x, dx')$. Such *i.i.d.* samples can be obtained by running the Markov chain until it has mixed so that $x_n \sim p$, collecting a couple $(x_n, x_n')$, and restarting. With $A_n := \gamma\Phi(x_n) \otimes \Phi(x_n') - \Phi(x_n) \otimes \Phi(x_n)$ and $b_n := r(x_n)\Phi(x_n)$, we study the recursion:

$$V_n = V_{n-1} + \rho_n((A_n - \lambda I)V_{n-1} + b_n). \tag{21}$$

In particular, $\mathbb{E}_q[A_n] = A$, $\mathbb{E}_p[b_n] = b$, and $A_n$ and $b_n$ are independent of the past $(V_k)_{k<n}$. For $\beta \in \{0, 1\}$, let $W_n^\beta := \|\Sigma^{-\beta/2}(V_n - V_\lambda^*)\|_{\mathcal{H}}^2$. Adapting the proof of Lemma 3, we exhibit a similar decreasing behavior of $W_n^0$ in expectation, hence showing that $\mathbb{E}[\|V_n - V_\lambda^*\|_{\mathcal{H}}^2] \to 0$ for well-chosen step sizes $\rho_n$. Finally, $\lambda$ is chosen to balance $\mathbb{E}[\|V_n - V_\lambda^*\|_{L^2(p)}^2]$ and $\|V_\lambda^* - V^*\|_{L^2(p)}^2$. We define $V_n^{(e)}$ and $V_n^{(t)}$ as the exponentially-weighted and the tail-averaged $n$-th iterates respectively:

$$V_n^{(e)} := \frac{\sum_{k=1}^n (1-\rho\lambda)^{n-k}V_{k-1}}{\sum_{k=1}^n (1-\rho\lambda)^{n-k}} \quad \text{and} \quad V_n^{(t)} := \frac{1}{n - \lfloor n/2 \rfloor + 1}\sum_{k=\lfloor n/2 \rfloor}^n V_{k-1}. \tag{22}$$

**Theorem 1.** *Let $n \geq 9$. Under assumption (A2) with $-1 < \theta \leq 1$, there exist a positive real number $\underline{\lambda}_\theta$ independent of $n$ such that, for $\lambda_0 \geq \underline{\lambda}_\theta$,*

(a) *Using $\lambda = \lambda_0 n^{-\frac{1}{3+\theta}}$ and a constant step size $\rho = \frac{\log n}{\lambda n}$, then:*

$$\mathbb{E}[\|V_n - V^*\|_{L^2(p)}^2] = O((\log n)n^{-\frac{1+\theta}{3+\theta}}).$$

*(b) Using $\lambda = \lambda_0 n^{-\frac{1}{2+\theta}}$ and a constant step size $\rho = \frac{\log n}{\lambda n}$, then:*

$$\mathbb{E}[\|V_n^{(e)} - V^*\|_{L^2(p)}^2] = O((\log n)n^{-\frac{1+\theta}{2+\theta}}).$$

*(c) Using $\lambda = \lambda_0 n^{-\frac{1}{2+\theta}}$ and a constant step size $\rho = \frac{2\log n}{\lambda n}$ for the first $\lfloor n/2 \rfloor - 1$ iterates and then a decreasing step size $\rho_k = \frac{1}{\lambda k}$, then:*

$$\mathbb{E}[\|V_n^{(t)} - V^*\|_{L^2(p)}^2] = O((\log n)n^{-\frac{1+\theta}{2+\theta}}).$$

A similar exponentially-weighted averaging scheme as in (b) has been used to study constant step size SGD in [28]. When $\gamma = 0$, the rates can be compared to existing results for SGD. For example, for $\theta \in [0,1]$, [65] proves almost sure convergence for regularized least-mean-squares without averaging at rate $O(n^{-\frac{1+\theta}{2+\theta}})$. The dependence in $\theta$ is similar to what we obtain. Moreover, under assumption (A1), we recover the same $O(1/\sqrt{n})$ convergence rate as [10] (see Prop. 4 stated in App. A.3). Finally, our bounds have a polynomial dependence in the horizon $1/(1-\gamma)$ of the MRP.

**Remark.** Because the Bellman operator is also a contraction mapping in $L^\infty$-norm, this analysis in $L^2$-norm might be adapted to the $L^\infty$-norm, using a modified Lyapunov function to study the ODE, *e.g.*, following [14]. The stochastic case could be handled using the smoothing technique recently developed by [20].

## 5 Stochastic TD with Markovian sampling

We now consider the truly *online* TD algorithm, where the samples are produced by a Markov chain. In particular, there is now a correlation between the current samples $(x_n, x'_n)$ and the previous iterate $V_{n-1}$. To control it, we assume that the Markov chain mixes at uniform geometric rate:

**(A3)** $\qquad \exists m > 0, \; \mu \in (0,1) \;\; \text{s.t.} \;\; \sup_{x \in \mathcal{X}} d_{TV}\left(\mathbb{P}(x_n \in \cdot | x_0 = x), p\right) \le m\mu^n, \qquad (23)$

where $d_{TV}$ denotes the total variation distance. This is always verified for irreducible, aperiodic finite Markov chains [46]. Note that the uniform mixing assumption might be relaxed by a weaker drift condition using the technique developed by [34] for linear TD, although its extension to the infinite-dimensional setting is not straightforward, and out of our scope. We give an example of continuous-state Markov chain with geometric mixing in Sec. 6. Furthermore, following [10], in our analysis we need to control the magnitude of the iterates almost surely. To do so, we add a projection step at each TD iteration:

$$V_n = \Pi_B[V_{n-1} + \rho_n((A_n - \lambda I)V_{n-1} + b_n)], \qquad (24)$$

where $\Pi_B$ is the projection on the $\mathcal{H}$ ball of radius $B > 0$. If $\|V_\lambda^*\|_\mathcal{H} \le B$, the convergence of the method is preserved. In the following theorem, we consider two regimes with different rates of convergence. In the first one, we assume like [10] that we are given an oracle $B$ upper-bounding $\|V_\lambda^*\|_\mathcal{H}$, with $B$ independent of $\lambda$. In the second one, we use the bound of Prop. 1, but this will affect the convergence rate since in this case $B = O(1/\lambda)$.

**Theorem 2.** *Assuming (A2) and that the samples are produced by a Markov chain with uniform geometric mixing (A3), the projected TD iterations (24) are such that:*

*(i) Using $\lambda = n^{-\frac{1}{2+\theta}}$, a constant step size $\rho = \frac{\log n}{2\lambda n}$, and using a projection radius $B$ independent of $\lambda$ provided by an oracle and such that $\|V_\lambda^*\|_\mathcal{H} \le B$, then:*

$$\mathbb{E}\left[\|V_n^{(e)} - V^*\|_{L^2(p)}^2\right] \le O\Big(\frac{(\log n)^2 n^{-\frac{1+\theta}{2+\theta}}}{\log(1/\mu)}\Big). \qquad (25)$$

*(ii) Using $\lambda = n^{-\frac{1}{4+\theta}}$, $\rho = \frac{\log n}{2\lambda n}$, and the projection radius $B = \sqrt{M_\mathcal{H}}\|r\|_{L^2(p)}/\lambda$, then:*

$$\mathbb{E}\left[\|V_n^{(e)} - V^*\|_{L^2(p)}^2\right] \le O\Big(\frac{(\log n)^2 n^{-\frac{1+\theta}{4+\theta}}}{\log(1/\mu)}\Big), \qquad (26)$$

*with $V_n^{(e)} = \sum_{k=1}^n (1-2\rho\lambda)^{n-k} V_{k-1} / \sum_{j=1}^n (1-2\rho\lambda)^{n-j}$.*

When an oracle is given for $B$ (i.e., setting (i)), we recover the same rate as *i.i.d.* sampling, up to a multiplicative factor $\log(n)/\log(1/\mu)$ which represents the mixing time of the Markov chain. If no oracle is provided (i.e., setting (ii)), the convergence will be slower because the bound $B$ is of order $1/\lambda$. Note that the slight changes in the definitions of $\rho$, $\lambda$, $V^{(e)}$, and the absence of constraint on $\lambda$, as compared to Thm. 1, are implied by the boundedness of the iterates. Following a similar study for SGD [50], we might compare these rates to those of a naive algorithm which we call "$\tau$-Skip-TD", for some $\tau \geq 1$, where only one every $\tau$ samples from the Markov chain is used to make TD updates:

$$V_n = \Pi_B[V_{n-1} + \rho_n((A_{n\tau} - \lambda I)V_{n-1} + b_{n\tau})], \tag{27}$$

For $\tau$ large enough, of the order of the mixing time of the Markov chain, the new sample $(x_{n\tau}, x'_{n\tau})$ is almost independent from the past ones $(x_{k\tau}, x'_{k\tau})_{k<n}$. Of course, since we need to generate $\tau$ times more samples to make a step, we must look at the distance of $V_{n/\tau}$ to the optimum. Such convergence rates for $\tau$-Skip-TD are derived in App. A.4, Cor. 2. In setting *(i)*, they are similar to Theorem 2 up to a $\log(n)$ factor. This suggests that making updates at each sample of the Markov chain is not more efficient than $\tau$-Skip-TD for large $\tau$, at least in our theoretical analysis. In practice, using all samples seems slightly better, especially for a slowly mixing Markov chain (see App.B.3). In setting *(ii)*, we obtain a rate for Skip-TD whose leading term does not depend on $\log(1/\mu)$ – which only appears in higher order terms – suggesting that the rate and parameters of Thm. 2, setting *(ii)* might be suboptimal.

**Remark.** The analysis of TD with linear function approximation by [60] does not require a projection step. Hence the necessity of the projection step might only be an artifact of our proof technique inspired by [10]. In the above experiments, we simply omit the projection step.

# 6 Experiment on artificial data

**Building a value function.** We build a toy model for which the main parameters can be computed in closed form. We consider the dynamics on the circle $\mathcal{X} = [0, 1]$ defined by: with probability $\varepsilon$, $x_{n+1} \sim \mathcal{U}([0, 1])$, and with probability $1 - \varepsilon$, $x_{n+1} = x_n$. Because the Markov kernel is symmetric, the invariant distribution is $p = \mathcal{U}([0, 1])$. In particular, the mixing parameter can be bounded explicitly with $m = 1$ and $\mu = 1 - \varepsilon$ (see App. B.1). Also, simple computations show that $V^*$ is an affine transform of $r$: $V^*(x) = ar(x) + b$, with $a = (1 - \gamma(1 - \varepsilon))^{-1}$ and $b = -a \int_0^1 r(u)du$. Hence we can build a $V^*$ with a given regularity by choosing an appropriate reward with the same regularity. We consider two rewards: $r_{\text{abs}}(x) := 2|x - 1/2|$ and $r_{\cos}(x) := (1 + \cos(2\pi x))/2$.

**Kernels on the torus.** We consider the RKHS of splines on the circle [67] of regularity $s \in \mathbb{N}^*$, denoted by $H^s_{\text{per}}$. It is a Sobolev space equipped with the following norm:

$$\|f\|^2_{H^s_{\text{per}}} = \left(\int_0^1 f(x)dx\right)^2 + \frac{1}{(2\pi)^{2s}} \int_0^1 |f^{(s)}(x)|^2dx. \tag{28}$$

Its corresponding reproducing kernel $K_s$ is a translation-invariant kernel defined by:

$$K_s(x, y) = 1 + (-1)^{s-1}\frac{(2\pi)^{2s}}{(2s)!}B_{2s}(\{x - y\}), \tag{29}$$

where $\{x\} := x - \lfloor x \rfloor$ and $B_j$ is the $j$-th Bernoulli polynomial [52]. Let us recall that the Fourier series expansion on the torus of a 1-periodic function $f \in L^2(p)$ is: $f(x) = \sum_{\omega \in \mathbb{Z}} e^{2i\omega\pi x}\hat{f}_\omega$, with $\hat{f}_\omega := \int_0^1 f(x)e^{-2i\omega\pi x}dx$, for $\omega \in \mathbb{Z}$. The kernel $K_s$ has an embedding in the space of Fourier coefficients $\Phi(x) = (\sqrt{c_\omega}e^{2i\omega\pi x})^\top_{m\in\mathbb{Z}}$ with $c_\omega := |\omega|^{-2s}$ if $\omega \neq 0$ and $c_0 := 1$. Using Parseval's theorem in Eqn. (28), one can compute the norm of $f$ from its Fourier coefficients $\|f\|^2_{H^s_{\text{per}}} = \sum_{\omega\in\mathbb{Z}} |\hat{f}_\omega|^2/c_\omega$. The operators $\Sigma$ and $\Sigma_1$ can be represented as countably infinite-dimensional matrices $\Sigma = \text{diag}(c)$ and $\Sigma_1 = (1 - \varepsilon)\Sigma + \varepsilon\sqrt{c}(\sqrt{c})^\top$. Hence the source condition states that $|\hat{f}_0|^2 + \sum_{\omega\neq 0} |\omega|^{2s(1+\theta)}|\hat{f}_\omega|^2 < \infty$. In particular, it holds if $f \in H^{s'}_{\text{per}}$, for any $s' \geq s(1+\theta)$. In our example, we consider two Sobolev spaces $H^1_{\text{per}}$ and $H^2_{\text{per}}$, and our two example functions have Fourier coefficients $(\hat{r}_{\text{abs}})_\omega = \frac{1-(-1)^\omega}{\pi^2\omega^2}$ for $\omega \neq 0$, and $(\hat{r}_{\cos})_\omega = 0$ for $|\omega| > 1$. The largest $\theta \in [0, 1]$ such that the source condition holds are indicated in the first row of Tab. 1.

**Results.** We run TD on functions $r_{\text{abs}}$ and $r_{\text{cos}}$, with kernels $K_1$ and $K_2$. We use parameters $\lambda$ and $\rho$ and exponential averaging as prescribed in Thm. 1 *(b)*. Each experiment is repeated 10 times and we record the mean $\pm$ one standard deviation. The implementation is based on a finite dimensional representation of the iterates $(V_k)_{k \leq n}$ in $\mathbb{R}^n$ (see further details in App. B.2). This implies computing the kernel matrix in $O(n^2)$ operations. To accelerate this computation when the eigenvalues decrease fast, we approximate it with the incomplete Cholesky decomposition [3]. In Tab. 1, we set $\varepsilon = 0.8$, $\gamma = 0.5$ and report the observed convergence rates *v.s.* the ones expected by Thm. 2, which are fairly consistent. In Fig. 1, we show the respective effects of varying $\varepsilon$ and $\gamma$. Larger values of $\varepsilon$ or $\gamma$ make the problem more difficult and slow down convergence, presumably in the constants without affecting the rates, as predicted by Thm. 2. Additional experiments are provided in App. B.3.

Table 1: Predicted and observed convergence rates with different reward functions and kernels.

|  | Sobolev kernel $s = 1$ | | Sobolev kernel $s = 2$ | |
|---|---|---|---|---|
|  | $r = r_{\text{abs}}$ | $r = r_{\text{cos}}$ | $r = r_{\text{abs}}$ | $r = r_{\text{cos}}$ |
| Maximal $\theta$ | $1/2$ | $1$ | $-1/4$ | $1$ |
| Predicted rate | $-0.6$ | $-0.67$ | $-0.43$ | $-0.67$ |
| Observed rate | $-0.72$ | $-0.64$ | $-0.58$ | $-0.64$ |

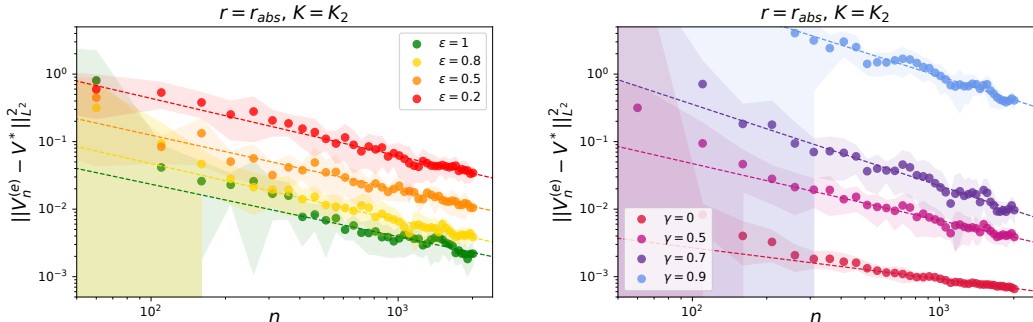

Figure 1: Respective effects of varying $\varepsilon$ (for $\gamma = 0.5$ fixed) and $\gamma$ (for $\varepsilon = 0.8$ fixed).

# 7 Conclusion

We have provided convergence rates for the regularized non-parametric TD algorithm in the *i.i.d.* and Markovian sampling settings. The rates depend on a source condition that quantifies the relative regularity of the optimal value function to the RKHS. They are compatible with our empirical observations on a one-dimensional MRP, but we have not proved optimality of such rates. Interesting directions include the extension to the TD($\lambda$) algorithm, which we believe can be achieved with similar tools, as well as more challenging extensions to control counterparts of TD (Q-learning, SARSA,...) for which the policy is optimized.

# Acknowledgements

The authors thank the anonymous reviewers and Pierre Marion for suggesting additional references. This work was supported by the Direction Générale de l'Armement, and by the French government under management of Agence Nationale de la Recherche as part of the "Investissements d'avenir" program, reference ANR-19-P3IA-0001 (PRAIRIE 3IA Institute). We also acknowledge support from the European Research Council (grant SEQUOIA 724063).

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
