# OpenReview forum: "A Non-asymptotic Analysis of Non-parametric Temporal-Difference Learning"
_NeurIPS.cc/2022/Conference — NeurIPS 2022 Accept_

### Official Review · Reviewer_XY19 · 2022-07-10

**Rating:** 5
**Confidence:** 3
**Soundness:** 3 good
**Presentation:** 4 excellent
**Contribution:** 3 good

**Summary:**

This paper focuses on solving the policy evaluation problem in reinforcement learning using non-parametric TD-learning. By introducing a regularization parameter $\lambda$, the authors derive (1) the convergence rate of the associated ODE, and (2) the convergence rate of non-parametric TD-learning under either i.i.d. or Markovian sampling. Numerical experiments agree with the theoretical findings.

**Questions:**

My main concerns are the need for the regularizer and the technical novelty compared to existing literature studying kernel-based RL.

**Limitations:**

This paper does not have any potential negative societal impact.

**Strengths And Weaknesses:**

This paper is well organized and well written. The authors start with the ODE associated with the deterministic variant of TD-learning, and use the Lyapunov function there to study the non-parametric stochastic TD-learning algorithm. While this is a highly technical paper, the structure makes the paper easy to follow.

Major Comments:

(1) What is the motivation of introducing the regularizer $\lambda$? Is it because $\Sigma$ is not necessarily invertible but $\Sigma+\lambda I$ is guaranteed to be invertible? The Bellman operator is also a contraction with respect to the $\ell_\infty$-norm, regardless of whether the Markov chain has a unique stationary distribution or not. Introducing the regularization parameter is to some extend equivalent to solving a different MDP with a smaller discount factor, and I feel it should be avoided if possible.

The TD-learning algorithm with the regularizer $\lambda$ is different than the original one. What is the updated algorithm? I do not think it is simply subtracting another $\lambda V_{n-1} (x_n)$ in the temporal difference.

(2) Regarding the convergence rate of the ODE. Suppose we exploit the $\ell_\infty$-norm contraction instead of the weighted $L_2$-norm contraction. Do we always get geometric convergence?

(3) The need for projection in TD-learning with Markovian sampling is somewhat problematic. First of all, assuming there is an oracle that gives the right projection set is not realistic. Second, the estimate of the size of the projection set depends on unknown parameters of the MDP. In [55], the authors provide a way of analyzing TD-learning with linear function approximation without a projection. Is it possible to remove the projection using similar techniques?

(4) The authors listed existing literature studying non-parametric RL, but did not compare the results and the techniques with them. I briefly checked [30], which is not for fitted Q-iteration (as claimed in this paper), but studies kernel based non-parametric LSTD and seems to be closely related to this work. The technical novelty is not entirely clear in the current manuscript.

Minor Comments:

(1) "Deadly Triad" refers to bootstrapping, off-policy, and function approximation (which does not have to be nonlinear). This should be made clear in the paper.

(2) TD-learning with either tabular representation or linear function approximation has been studied extensively in the literature, and the convergence rate there is $1/\sqrt{n}$. Some discussion seems needed to clarify why non-parametric TD has a slower convergence rate.

---

> ### Author Response · Authors · 2022-08-01
> **Various clarifications and correction of a reference**
>
> Major Comments:
>
> (1) The regularizer indeed ensures that $\Sigma + \lambda I$ is invertible. We analyze the regularized TD learning algorithm (12) because it allows studying to case where $||V^*||_H$ is infinite, i.e., $V^*$ does not belong to the RKHS. If $V^* \in H$, the regularization is not necessary, as seen in Prop.4 in App.A.3.
>
> We politely disagree with the statement: "Introducing the regularization parameter is to some extend equivalent to solving a different MDP with a smaller discount factor". This is not equivalent to changing the discount factor.
>
> The updated (regularized) algorithm is written on eqn. (21). It amounts to subtracting $\rho_n \lambda V_{n-1}$. This term is not inside the temporal-difference because it does not depend on $\Phi(x_n)$. This is indeed different from subtracting $\lambda V_{n-1}(x_n)$ in the temporal difference (which we agree would be like changing the discount factor).
>
> (2) We agree that the Bellman operator is also a contraction mapping with respect to the $l^\infty$-norm. However, we are not sure that the same proof extends to this norm for studying the ODE. First, there are technical difficulties due to the non-differentiability of this norm when we compute the derivative of the Lyapunov function. In general, the descent lemma does not hold. Furthermore, the convergence of $V^*_\lambda$ to $V^*$ in Prop.2 is specific to the $L^2$ norm (although it can be extended to intermediate norms between the $L^2$ norm and the $H$ norm). We do not think that a similar result exists in $l^\infty$ norm. Overall, an $l^\infty$-norm analysis could possibly be done, but it cannot be deduced directly from our analysis. Asymptotic stochastic approximation results in $l^\infty$-norm exist e.g. in [Bertsekas & Tsitsiklis, 1995, Neuro-dynamic programming: an overview, Prop. 4.4]. We would like to stress that former asymptotic and non-asymptotic analyses of TD are also in $L^2$ norm (see e.g., [61]).
>
> (3) We agree with Reviewer XY19 about his or her remarks on the projection step. The analysis of linear TD by [55] indeed removes the projection step. We agree that this would be a desirable feature in our case as well. We have chosen to follow the general scheme of the analysis of [10] rather than [55], because it establishes finite time bounds that are *independent of the conditioning* of the full-rank feature covariance matrix $\Sigma$. This is a desirable property for our non-parametric analysis because our infinite-dimensional operator $\Sigma$  is not invertible (and cannot have full rank).
> It could be interesting to investigate in future work if a similar analysis to [55] using Steiner's method can be carried out in our case to remove the projection step. Note that in the experiments, we did not have to use any projections.
>
> (4) We have made the following mistake in the manuscript: references [30] and [42] have been permuted: [30] indeed refers to LSTD, and [42] to fitted Q iteration. We are sorry about that and we have updated the manuscript to correct this mistake. Kernel-based LSTD is indeed related to this work. In particular, the operators $\Sigma$ and $\Sigma_1$ are also used in [30], and the eigenvalues of the kernel are also used. However, the scope of [30] is a bit different: they study the statistical (or estimation) error of an offline algorithm (LSTD), whereas we analyze the convergence of an online algorithm (TD). Overall, we view our analysis as complementary to [30], just like studying batch least-squares and SGD on least-squares are related, yet different problems.
>
> Minor Comments:
>
> (1) We agree that the "Deadly Triad" refers to function approximation in general. We have clarified that in the manuscript.
>
> (2) We agree that TD-learning with either tabular representation or linear function approximation has convergence $1/\sqrt n$. This is something we recover in Prop. 4 in Appendix A.3. We obtain the same rates for non-parametric TD when $\theta=0$ (the rate being $n^{-(1+\theta)/(2+\theta)}$. The case $\theta=0$ corresponds to $V^* \in H$, which is  enforced by the minimal projected Bellman error solution for linear function approximation. We mention this in particular on lines 244-245: in this case, regularization is not necessary. Furthermore, we would like to stress that the rates under assumption (A2) can be slower, identical or faster than $1/\sqrt n$ depending on the source condition. In particular, if $\theta >0$, the obtained rates are faster than $1/\sqrt n$, and can be as fast as $n^{-2/3}$.

---

> > ### Comment · Reviewer_XY19 · 2022-08-09
> > **Response to the authors**
> >
> > I thank the authors for the detailed response. Some of my concerns are still not fully addressed and I intend to keep my score.
> >
> > I understand that you are subtracting $\lambda V_{n-1}$ in the entire update as opposed to just inside the temporal difference. However, I believe introducing $\lambda$ this way also has some impact on the effective discount factor of the problem. Since Proposition 2 provides an upper bound on the difference, I think this is ok and am satisfied with the authors’ response on this point.
> >
> > There are some papers that address the difficulty of dealing with algorithms under $\ell_\infty$-norm contraction operators. For example, the authors of [] have shown that with some additional effort the norm-square function can also be used as a Lyapunov function to study the associated ODE. As for the stochastic algorithm, [] introduced a smooth version of sup-norm square so that the decent lemma holds with the new potential function.
> >
> > [1] Borkar, V. S., & Soumyanatha, K. (1997). An analog scheme for fixed point computation. i. theory. IEEE Transactions on Circuits and Systems I: Fundamental Theory and Applications, 44(4), 351-355.
> >
> > [2] Chen, Z., Maguluri, S. T., Shakkottai, S., & Shanmugam, K. (2020). Finite-sample analysis of contractive stochastic approximation using smooth convex envelopes. Advances in Neural Information Processing Systems, 33, 8223-8234.
> >
> > I understand that [30] studies off-line LSTD while this paper studies the online version. However, given that on-line and off-line TD in the tabular setting and under linear function approximation are well-studied in the literature, and [30] studied off-line non-parametric TD, the challenges in extending the results to online nonparametric TD are still not entirely clear. What is the major technical challenge in this work and what is the novel idea that is proposed to overcome the challenge?

---

> > > ### Author Response · Authors · 2022-08-09
> > > **Response to reviewer XY19**
> > >
> > > We would like to thank Reviewer XY19 for his or her further comments.
> > >
> > > Concerning the $\ell_\infty$-norm analysis, given the further references that you provided on stochastic approximation, we indeed believe that the analysis could be extended to this change of norm. We will add a comment on this, and cite the additional references.
> > >
> > > Concerning the link with [30], we would like to insist on the fact that, although they are related, LSTD and TD are different algorithms. TD is an instance of stochastic approximation, while LSTD is not. Therefore, the analyses of the algorithms are quite different in nature, and in particular, the finite-time analysis of TD cannot be directly deduced from the analysis of LSTD (even in the linear approximation setting). For instance, [30] mainly employ *statistical tools* (like Rademacher complexities), while we use tools from *stochastic approximation* or stochastic optimization. These are two different approaches to two different, but related, problems. Therefore, some objects will appear in both analyses: the covariance operator, the horizon of the MRP, the richness of the RKHS and the complexity of the optimal value function (also called the capacity and source condition in optimization),... Overall, to give an analogy in optimization, the difference between LSTD and TD is the same as between linear least squares (operating a system inversion) and the least-mean-square algorithm (an instance of SGD on the least squares objective). This is the same in non-parametric settings, but to the best of our knowledge, an analysis of non-parametric least-mean-square (like the one of [28]) is not a direct consequence of the statistical study of kernel least squares.
> > >
> > > The major technical challenges in this work are twofold:
> > > - one is to deal with a non-invertible covariance operator $\Sigma$. This did not occur in the linear approximation setting, e.g., in [10] or [55] who assumed that $\Sigma$ has full rank (which we do not assume here). We deal with this issue by introducing regularization.
> > > - the second one is to prove the convergence of TD to $V^*$, even when $V^* \notin H$. This is different from previous analyses which study convergence to the fixed point of the projected Bellman operator, including [30]. This is allowed by Propositions 1 and 2, which extend similar result coming from the study of non-parametric least-mean-square, to temporal differences. Furthermore, this directly provides rates that are adaptive to the regularity of $V^*$, which we believe is novel for TD.

---

### Official Review · Reviewer_3hWK · 2022-07-11

**Rating:** 5
**Confidence:** 4
**Soundness:** 3 good
**Presentation:** 2 fair
**Contribution:** 2 fair

**Summary:**

The paper studies the temporal difference algorithm for estimating the value function of a Markov decision process, assuming that the control policy is already applied and the resulting Markov chain is homogenous and stationary. Under certain assumptions, the rates of convergence of the weighted averages of the value functions provided by a regularized temporal difference algorithm are shown, averaging over the stochasticity, as well as the state space.

**Questions:**

I would like the authors to address the points discussed under Weaknesses, and also explain how they can improve the presentation. The latter seems necessary as there are unclarity and lack of enough explanation in some places.

In the abstract, 'source condition' is unclear.
98: 'but ...' makes ambiguity.
106: define a 'nonnegative' reward.
In (1), it is unclear what is known and what is not. The main interest is in the case that the Markov transitions are unknown, but from the rest of the presentation, it does not seem to be the case as some of the quantities need the transition to be computed. Note that as the processes is assumed to be stationary and/or mixed, the authors need to argue why the known distribution does not provide the transition.
Harris ergodicity, especially its regeneration set, need to be defined, and the discussion in these lines need more explanations.
132: it is unclear what is n, and how these computations relate to the setting.
Intuitions of Lemma 2 that the operator is like a contraction are required, as well as implications of such a fact.
The first paragraph of Section 3 is unclear.
On proposition 1, I think we are interested in the solutions of (12), and not those of (13).
I am not convinced about necessity and/or usefulness of regularizing the TD.

**Limitations:**

Mentioned in Weaknesses.

**Strengths And Weaknesses:**

Strength:
The framework seems technically solid. The presented results are explained rigorously, and the setting is abstract enough to include non-tabular Markov decision processes, as long as the feature space is an RKHS.

Weaknesses:
The setting is a little artificial.
The experiments are not clear to be generalizable.
The presentation is too compact and a little hard to follow, and is also unclear in some places.
Importance of the problem is not sufficiently motivated as the policy is already applied, the transitions are mixed and have reached to the stationarity, and now the goal is to find the value function. This, as well as the next, restrict the applicability of the proposed approach for being used as the evaluation step of a reinforcement learning policy.
The results are in the form of average-case analysis. Intuitively, that means that if we repeat the setting many times, we can learn the value function nicely. However, the more interesting analysis is one that can establish accuracy of the learned value function based on a single trajectory. So, in some sense, the policy evaluation is analyzed in an offline fashion, while for offline reinforcement learning policies, evaluations are not the main obstacle.
Finally, I do not think that Section 4 fits well in the framework as restarting for many times, together with the fact that the expected learning error is studied, defeat the purpose and limit the practicality of the approach.

---

> ### Author Response · Authors · 2022-08-01
> **Various clarifications**
>
> (1) In this paper, we study the *policy evaluation* problem, and in particular the TD(0) algorithm. Policy evaluation is a subproblem of reinforcement learning, but a significant one. In particular, even though the policy is fixed, the convergence of TD(0) with function approximation is a non-trivial problem. As noted in the introduction, TD(0) with linear function approximation only converges to the minimizer of the projected Bellman error. We propose an algorithm that generically converges to the value function.
> As noted by [10], the assumption that the Markov chain begins in steady-state is not essential (if it has not mixed yet, we could apply our analysis after the chain has approximately mixed), but it simplifies the presentation.
>
> (2) We do not mean the experiments to be generalizable nor to demonstrate the interest of the algorithm on real problem instances. However, the experiments are here to illustrate the theorems and give an intuition of the effect of the different parameters.
>
> (3) We agree that our convergence results are in the form of average-case analysis. Stochastic approximation theory could also provide almost sure convergence results, but they are mostly *asymptotic* convergence results. Instead, we follow a line of work which provides average-case analyses for TD learning, such as [10], [55], [61], [64], [40]... Average-case analysis are also common in the study of the convergence rates of stochastic gradient descent (see e.g., [28], [49], [9]). Furthermore, we would like to stress that in the experiments (see Figure 1), we plot the mean +- one standard deviation over 10 runs. Each run is one instance of the TD algorithm, without any restarting. Even though there are some unavoidable random fluctuations, the average-case analysis seems to be relevant.
>
> (4) We agree that the i.i.d. scenario of Sec.4 is not realistic in RL applications. This is why we study in Sec.5 the Markovian setting. We merely begin with the i.i.d. case as an intermediate step to make our analysis more progressive.
>
> (5) In the abstract, although its formal definition is not directly written, the "source condition" is followed by the brief description: "relating the regularity of the optimal value function to the RKHS". We give the formal definition of the source condition on page 5, line 179. We would like to stress that the source condition is an existing terminology which has already been used for the study of stochastic gradient descent (see [27], [9] and references therein).
>
> (6) In Eqn. (1), $\gamma$ is supposed to be known, but the reward function $r$ and the transition probabilities in the expectation are unknown. TD learning is indeed useful in the case where $r$ and the transitions are unknown (else one could use e.g., value iteration). We consider this framework (reward and transitions unknown) throughout the paper. We do not use these transitions to compute the TD updates in eqn. (4). The transitions need to be known if we wanted to run the population version of TD in eqn. (12). However, this algorithm is never run in practice, and we only consider it as an intermediate step of our analysis. This is not the algorithm analyzed in Thms 1 and 2.
>
> (7) Once the process has mixed, the samples are drawn according to the invariant distribution $p$. This distribution does not need to be known to run TD. We could imagine to learn it by observing the samples, but this would not be of any help to run TD. We do not think that, even knowing $p$, we could recover the transition probabilities. For example, in Sec.6, the invariant distribution $p$ is uniform, but this brings no information on the transition kernel.
>
> (8) We analyze the regularized TD learning algorithm (12) because it allows studying to case where $||V^* ||_H$ is infinite, i.e., $V^*$ does not belong to the RKHS. If $V^* \in H$, the regularization is not necessary, as seen in Prop.4 in App.A.3.
>
> (9) Other clarifications:
> + l. 98: the word "but" has been removed.
>
> + l. 106: the reward is not required to be non-negative in our analysis. More generally, there is no obstacle to having negative rewards in a discounted MRP.
>
> + For Harris ergodicity and regeneration sets, we already refer to [1],[31]. Since the definitions are rather technical and outside of the scope of the paper, we have prefered to refer to the references. However, in the camera-ready version of the paper, since we have more room we could add more details.
>
> + l. 132: $n$ refers to the index of the update (4) just above. The $n$-th update has complexity $O(n^2)$. More details on the implementation are given in Appendix B.2.
>
> + The intuition of Lemma 2 is given in the paragraph below (l. 193 -- 197). In a simplified setting, Lemma 1 and Lemma 2 are equivalent. The intuition of Lemma 1 is given on l. 167 -- 172. The main implication of Lemma 2 is to prove Lemma 3. We have added a mention of that on line 235.
>
> + In Prop. 1, we are interested in the solution of (13) (the fixed-point of (12)).

---

### Official Review · Reviewer_KMJg · 2022-07-11

**Rating:** 6
**Confidence:** 4
**Soundness:** 3 good
**Presentation:** 3 good
**Contribution:** 3 good

**Summary:**

This paper analyzes the kernel based (on-policy) TD learning. Specially, they consider the case where TD learning is performed with the value function in a reproducing kernel Hilbert space (RKHS) (eq 3, 4). They provide convergence guarantees when the true value function V* does not belong to the RKHS under a so-called source condition (A2). They also provided non-asymptotic convergence rate for the algorithm under i.i.d setting and the Markovian setting (where the state action sequence is sampled from a fixed policy).


**Questions:**

The paper analyzed the so-called regularized TD learning (eq 12). My understanding is that this $\lambda$ is for the use of the representer theorem. But this parameter needs to be tuned in practices. In the main theorem (theorem 1), the analysis needs $\lambda > \lambda_{\theta}$. Is there any insight on the parameter $\lambda_{\theta}$ here, does it depend on $\gamma - 1$?

**Limitations:**

Yes

**Strengths And Weaknesses:**

Strength:
1. The paper is well written and structured. The authors gradually build the machinery from dynamic programming, RHKS, stochastic approximations (and etc) so as to introduce their main results and analysis.
2. As far as I know, the main technical contributions of the paper is the analysis of kernel based on-policy TD learning setting in the framework of RKHS. The general framework of analysis is similar from the analysis of TD-learning with linear function approximation, the results in terms of RKHS is technical.  To my knowledge, theoretical guarantees of TD-learning with general nonlinear function approximation is still lacking. Kernel based TD learning could be a step forward from existing TD-learning with linear function approximation.  And such results can provide more insight and support into those kernel based methods.

Weakness:
One potential drawback could be that the main framework of analysis bears some resemblance to the analysis of TD learning with linear function approximation. But again, performing such analysis using tools from RHKS is still technical.

Based on the results in ref[0], the projection step for the Markovian case for TD-learning may not be necessary.

ref:
[0] Finite-Time Error Bounds For Linear Stochastic Approximation and TD Learning, R. Srikant, Lei Ying.

---

> ### Author Response · Authors · 2022-08-01
> **Clarifications on the regularization parameter**
>
> We would like to thank Reviewer KMJg for the comments and questions raised. In the revised manuscript, we will add a mention to [Srikant2019] for possibly removing the projection step, and a short discussion on the dependence of $\lambda_\theta$ on $\gamma$. Please understand that we prefer not to add them to the manuscript at the moment because of the 9-page limit which will be extended to 10 for the camera-ready version.
>
> (1) The analysis of linear TD by [Srikant2019] indeed removes the projection step. We agree that this would be a desirable feature in our case as well. We have chosen to follow the general scheme of the analysis of Ref.[10] rather than [Srikant2019], because it establishes finite time bounds that are *independent of the conditioning* of the full-rank feature covariance matrix $\Sigma$. This is a desirable property for our non-parametric analysis because our infinite-dimensional operator $\Sigma$  is not invertible (and cannot have full rank).
> It could be interesting to investigate in future work if a similar analysis to [Srikant2019] using Steiner's method can be carried out in our case to remove the projection step. Note that in the experiments, we did not have to use any projections.
>
> (2) We analyze the regularized TD learning algorithm (eqn. 12) because it allows studying to case where $||V^* ||_H$ is infinite, i.e., $V^*$ does not belong to the RKHS. If $V^* \in H$, the regularization is not necessary, as seen in Prop.4 in Appendix A.3.
> The parameter $\lambda$ is not required for a "representer theorem" in a strict sense in our analysis. Choosing a regularization $\lambda >0$ ensures that the fixed point $V^*_\lambda$ of eqn. (13) is in the RKHS. Concerning the iterates, even for $\lambda=0$, it is straightforward from eqn. (12) that $\forall n, V_n \in \text{span}(\Phi(x_1),..., \Phi(x_n))$, which is a kind of representer theorem. This is discussed in particular in Appendix B.2.
>
> (3) In Thm. 1, we choose $\lambda$ as a constant $\lambda_0$ times a term decreasing with $n$. To obtain convergence, we need $\lambda$ to go to zero when $n$ goes to infinity. The dependence of $\lambda$ on $n$ is tuned to obtain the best rates in the theorems (given our analysis). The constant $\lambda_0$ must be chosen larger than a certain $\lambda_\theta$, of which we give an explicit expression in the proof (see pages 24-25). It behaves as a constant times $1/\bar \rho$, where $\bar \rho$ is defined in Lemma 6 (page 19). Overall, one must choose $\lambda_0 \geq \frac{c}{1-\gamma}$, where $c$ is a constant independent of $\gamma$. So this depends linearly on the horizon $\frac{1}{1-\gamma}$ of the MRP. Note that in the whole paper, we have not tried to optimize the dependences on $\frac{1}{1-\gamma}$, so it is possible that some of them are suboptimal. In practice, $\lambda_0$ can be tuned if $M_H$ is unknown, but we have not seen much influence of $\lambda_0$ in our experiments. We do not believe it to be a clear limitation, but probably only an artifact of the proof technique (for comparison, we do not need it in Thm.2).

---

### Official Review · Reviewer_15bu · 2022-07-19

**Rating:** 7
**Confidence:** 3
**Soundness:** 4 excellent
**Presentation:** 3 good
**Contribution:** 3 good

**Summary:**

This paper studies the convergence of the regularized non-parametric TD(0) algorithm with RKHS approximation. For both the IID and Markovian noise settings, convergence rate bounds have been obtained.  Numerical results have been given to support the theory.

**Questions:**

1. [Cai2019] has discussed some results for neural network approximation case. In the introduction, the authors mentioned that studying the RKHS case could bring us closer to understanding what happens with other universal approximators used in practice, like neural networks. Can the authors comment on the connections between their paper and [Cai2019]?

2. [Hu2019] has given some exact analytical formulas for the TD error for linear approximation on countable state space. Is it possible to obtain similar exact formulas for the RKHS approximation on general state case?

3. [Durmus2021] has addressed the TD error for linear approximation on general state space. It will be interesting to compare the assumptions in [Durmus2021] with the Harris ergodic assumption in this paper. In the linear approximation case, can the analysis method in this paper be used to get some improvements over [Durmus2021]?

**Limitations:**

Yes, the authors have discussed the limitations.

**Strengths And Weaknesses:**

This paper is quite original. The quality is good. The paper is also well written.

Strengths:
1. Originality: This paper is original. The analysis is new and novel.

2. Quality: The contributions are very solid. Theory for both IID and Markov noise cases have been discussed. Numerical results are also provided.

3. Clarity: The paper is well written.

4. Significance: The contributions are significant. The analysis is very interesting.

Weaknesses:
  The connections to the following relevant papers are missing, and some clarifications/discussions are needed.
[Cai2019] Q. Cai, Z. Yang, J.D. Lee, Z. Wang. Neural temporal-difference learning converges to global optima. NeurIPS2019.

[Hu2019] B Hu, U Syed. Characterizing the exact behaviors of temporal difference learning algorithms using Markov jump linear system theory. NeurIPS 2019.

[Durmus2021] A Durmus, E Moulines, A Naumov, S Samsonov, H Wai. On the stability of random matrix product with Markovian noise: Application to linear stochastic approximation and TD learning. COLT 2021.

Specific suggestions are given as follows:

1. [Cai2019] has discussed some results for neural network approximation case. In the introduction, the authors mentioned that studying the RKHS case could bring us closer to understanding what happens with other universal approximators used in practice, like neural networks. Hence it seems quite relevant to discuss the existing convergence theory for the neural network case.

2. [Hu2019] has given some exact analytical formulas for the TD error for linear approximation on countable state space (under both IID/Markov assumptions). Is it possible to obtain similar exact formulas for the RKHS approximation on general state case? Some discussion will be helpful.

3. [Durmus2021] has addressed the TD error for linear approximation on general state space. It will be interesting to compare the assumptions in [Durmus2021] with the Harris ergodic assumption in this paper. In the linear approximation case, can the analysis method in this paper be used to get some improvements over [Durmus2021]?

---

> ### Author Response · Authors · 2022-08-01
> **Comments on the additional references**
>
> We would like to thank Reviewer 15bu for the very interesting additional references that we were not aware of. In the revised manuscript, we will add a discussion on the given references, as detailed below. Please understand that we prefer not to add them to the manuscript at the moment because of the 9-page limit which will be extended to 10 for the camera-ready version.
>
> (1) [Cai2019] consider TD(0) with function approximation, using a one-hidden layer neural network with finite width. They prove $1/\sqrt n$ convergence to the solution with minimal projected Bellman error, which is an interesting extension of classical results for linear function approximation. However, in Prop. 4.7 of [Cai2019], there is still a projection error term which is equal to zero only if the value function is a function generated by a neural network. The authors mention that this function space "is a subset of an RKHS". Therefore, this essentially corresponds to the case $\theta \geq 0$ with our notations, i.e., when the value function is inside the RKHS. Our main focus in the paper is to prove convergence to $V^*$ in the case $\theta < 0$, i.e., when it is not inside the RKHS.
> More generally, it could be interesting in further work to investigate the connections between our RKHS framework, and the infinite-width limit of neural TD in [Cai2019]. Indeed, there is a parallel line of work studying the effect of stochastic gradient descent for optimizing one hidden-layer neural networks, and we have shown some similarity between SGD and TD.
>
> (2) The analysis of [Hu2019] characterizes the exact behaviour (hence providing both upper and lower bounds) of linear TD with finite state space. The framework of Markov jump linear system seems strongly linked to finite state spaces. In particular, the expressions of the transition probabilities appear explicitly in the analysis. They could maybe be replaced by more general expectations in the continuous state-space case, but this does not appear straightforward. This is an interesting direction but we believe it to be a bit outside of the focus of the current paper.
>
> (3) The recent results by [Durmus2021] provide finite bounds for linear stochastic approximation, which apply to linear TD. They relax the uniform mixing assumption that we use by replacing it by a weaker drift condition, and by removing a boundedness assumption. This appears as a promising direction, e.g., to remove the projection step that we have to use in the Markov setting. Yet, it is not straightforward to see whether this analysis designed for "random matrix products" can be directly extended to infinite-dimensional operators in the non-parametric case. After a quick look at the proofs, some constants (see e.g., Lemma 16 in [Durmus2021]) depend on the dimension $d$ of the linear approximation (typically infinite in the non-parametric case).

---

### Meta-Review · Area_Chair_SxaK · 2022-08-25

**Recommendation:** Accept
**Confidence:** Certain

**Metareview:**

The paper studies the convergence of non-parametric temporal-difference learning in the non-asymptotic regime. All referees agree that the paper is technical sound and the result is important to further our theoretical understanding of reinforcement learning. The paper merits acceptance to the conference.

**Award:**

No

---

### Decision · Program_Chairs · 2022-09-14

Accept